# An Efficient Hybrid RSS-AoA Localization for 3D Wireless Sensor Networks

**DOI:** 10.3390/s19092121

**Published:** 2019-05-07

**Authors:** Thu L. N. Nguyen, Tuan D. Vy, Yoan Shin

**Affiliations:** School of Electronic Engineering, Soongsil University, Seoul 06978, Korea; thunguyen@ssu.ac.kr (T.L.N.N.); tuanvyduc@ssu.ac.kr (T.D.V.)

**Keywords:** wireless sensor networks, hybrid localization, received signal strength, angle-of-arrival, suboptimal, weighted least squares estimate

## Abstract

Wireless sensor networks (WSNs) enable many applications such as intelligent control, prediction, tracking, and other communication network services, which are integrated into many technologies of the Internet-of-Things. The conventional localization frameworks may not function well in practical environments since they were designed either for two-dimensional space only, or have high computational costs, or are sensitive to measurement errors. In order to build an accurate and efficient localization scheme, we consider in this paper a hybrid received signal strength and angle-of-arrival localization in three-dimensional WSNs, where sensors are randomly deployed with the transmit power and the path loss exponent unknown. Moreover, in order to avoid the difficulty of solving the conventional maximum-likelihood estimator due to its non-convex and highly complex natures, we derive a weighted least squares estimate to estimate jointly the location of the unknown node and the two aforementioned channel components through some suitable approximations. Simulation results confirm the effectiveness of the proposed method.

## 1. Introduction

Localization plays a beneficial role in enabling various applications in wireless sensor networks (WSNs), especially for location-aware applications. By computing the locations of wireless nodes in the networks, it is possible to reveal the knowledge for the services that the users have used. Regarding the WSN characteristics (e.g., short-range communication, limited battery and capacity, etc.), one big challenge is to solve the localization problem with the constraints in energy consumption and communication bandwidth [1,2]. For example, developing a smart parking system for vehicles produces a 2D/3D plan to find the nearest parking slot. In order to delivery a better parking experience, other Internet-of-Things (IoT) functionalities such as mobile location service should be integrated to achieve a real-time availability status. Detailed surveys on location-based applications can be found in [3,4,5,6].

Recently, localization based on hybrid measurements has received much attention due to its accuracy and wide range of applications in WSNs such as surveillance, navigation, and tracking. A range-based hybrid scheme can be any combination of time-of-arrival (ToA), time-difference-of-arrival (TDoA), angle-of-arrival (AoA), and received signal strength (RSS) [7,8,9]. Among them, the hybrid RSS-AoA localization can significantly improve the estimation accuracy and have lower costs [10] because the AoA measurement can be adopted to achieve self-localization, while RSS measurement is obtained easily with low accuracy in harsh environments. For instance, it has been shown in [6] that a minimum of two AoA measurements from two different anchors is enough to obtain a location estimation of a target. Moreover, the hybrid RSS-AoA localization can avoid the strict requirement of precise timing synchronization in ToA and TDoA measurements. Thus, the localization problem with hybrid RSS-AoA measurements needs to be carefully investigated.

Despite the recent interest in the hybrid RSS-AoA approach, there remain two key challenges for practical implementation. First, it is assumed that the receiver is able to observe and extract hybrid measurement from the same received signal. However, this assumption may not hold in practice as practical sensors are not yet able to measure both measurements directly. Second, the conventional receiver architecture designed for RSS and AoA may operate with different power sensitivities at the receiver. Thus, for a system that involves hybrid measurements, the receiver architecture should be optimized according to a specific application purpose. Regarding the RSS, all sensor nodes are supposed to have radios. The distance information thus can be obtained from the RSS measurements. For the AoA, an antenna array [11,12,13,14,15] or an omnidirectional antenna [16,17] is implemented at the anchor nodes to measure AoA measurements from different normal sensors. In such settings, all normal sensors are equipped with a digital compass [18,19] to obtain the orientation information and to reduce energy consumption. When both the distance and the AoA measurements are available at each anchor node, the target location can be uniquely determined by using a certain number of anchors [11,12,13]. In order to quickly access range information, a quantized RSS method is implemented to estimate the distance. In summary, the main contributions of this paper are summarized as follows.
(i)We introduce a 3D hybrid localization that combines the RSS and the AoA measurements, where the channel parameters, i.e., the transmit power and the path loss exponent (PLE) are unknown.(ii)We propose a fast and accurate localization scheme where an unbiased solution is developed based on the hybrid measurements. In particular, we transform the original maximum likelihood estimator (MLE), which is non-convex and difficult to solve, into an alternative form after some suitable approximations, then apply a certain numerical optimization.(iii)We derive the Cramer–Rao lower bound (CRLB) and geometric dilution of precision (GDoP) analysis for the proposed weighted least squares (WLS).(iv)In the simulation results, we compare our work with other existing localization approaches under some likely practical conditions in terms of root mean squared error and bias.

The remainder of the paper is organized as follows. In Section 2, we introduce a 3D localization problem including the system description and related works. In Section 3, we present step-by-step procedures on our solution for improving the localization accuracy and reducing the run time. Simulation results are demonstrated in Section 4, followed by conclusions and future works in Section 5.

## 2. System Description and Related Works

### 2.1. Notations

The following notations are used throughout the paper. E(·) and Var(·) denote the expectation and the variance operators, respectively. We denote $A=[a_{ij}]$ as a matrix whose entries are $a_{ij}$, $I_{n}$ as the n×n identity matrix and 0 as all-zero vectors or matrices. ${\left(\cdot \right)}^{T}$ and ${\left(\cdot \right)}^{-1}$ correspondingly denote the transpose and the inverse operations. diag(·) denotes the diagonal matrix. $N(\mu ,\sigma^{2})$ denotes the Gaussian distribution with a mean μ and a variance $\sigma^{2}$. $\left|\right|\cdot {\left|\right|}_{1}$ and ||·|| denote the $\ell_{1}$ and the $\ell_{2}$-norm, respectively. Unif[a,b] denotes the uniform distribution on the interval [a,b]. ${\left[y\right]}_{\left[i:j\right]}$ denotes the ith to the jth elements of a vector y.

### 2.2. Problem Formulation

We assume there are *N* anchor nodes and one unknown node in a 3D WSN, where the locations of the anchor nodes are denoted by $a_{1},\cdots ,a_{N}\in R^{3}\;\left(a_{i}={\left[x_{i},y_{i},z_{i}\right]}^{T},i=1,\cdots ,N\right)$ and the location of the unknown node is denoted by $x={\left[x,y,z\right]}^{T}$, as illustrated in Figure 1. We also assume the anchors are equipped with hybrid RSS-AoA antennas. Thus, the combined measurements are given by: (1)$$\begin{array}{c} P_{i}=P_{0}-10\beta \log_{10}\frac{d_{i}}{d_{0}}+m_{i},\\ \phi_{i}=\tan^{-1}\left(\frac{y-y_{i}}{x-x_{i}}\right)+n_{i},\\ \alpha_{i}=\cos^{-1}\left(\frac{z-z_{i}}{d_{i}}\right)+v_{i}, \end{array}$$
where $P_{i}$ is the received signal power (in dB) of the signal transmitted by the target node at the ith anchor, $\phi_{i}$ and $\alpha_{i}$ are the azimuth and the elevation angles, respectively, $d_{i}=||x-a_{i}\left|\right|$ is the distance between the ith anchor node and the target node, $P_{0}$ is the power measured at a reference distance $d_{0}$, β is the PLE, and $m_{i}∼N\left(0,\sigma_{m_{i}}^{2}\right),n_{i}∼N\left(0,\sigma_{n_{i}}^{2}\right),v_{i}∼N\left(0,\sigma_{v_{i}}^{2}\right)$ represents the log-normal shadowing and the measurement errors for the azimuth and the elevation angles, respectively. Without loss of generality, we assume $d_{0}=1$ m. Note that $P_{0}$ is relevant to the transmit power $P_{T}$. Our goal is to determine x based on the measurements (Equation 1), where $P_{0}$ and β are treated as unknown parameters.

Given the observation vector $r={\left[P^{T},\phi^{T},\alpha^{T}\right]}^{T}\in R^{3N}$, where $P=\left[P_{i}\right],\phi =\left[\phi_{i}\right]$ and $\alpha \;=\;\left[\alpha_{i}\right]\;\left(i\;=\;1,\cdots ,N\right)$, the joint probability density function (PDF) is given as:(2)$$p\left(r|\theta \right)=\prod_{i=1}^{3N}\frac{1}{\sqrt{2\pi \sigma_{i}^{2}}}\exp\left\{-\frac{1}{2\sigma_{i}^{2}}{\left[r_{i}-f_{i}\left(\theta \right)\right]}^{2}\right\},$$
where $\theta \;=\;{\left[x,y,z,P_{0},\beta \right]}^{T}$, $f_{i}\left(\theta \right)=P_{0}-10\beta \log_{10}\frac{d_{i}}{d_{0}}$, $f_{i+N}\left(\theta \right)\;=\;\tan^{-1}\left(\frac{y-y_{i}}{x-x_{i}}\right)$, $f_{i+2N}\left(\theta \right)\;=\;\cos^{-1}\left(\frac{z-z_{i}}{d_{i}}\right)$, $\sigma_{i}^{2}\;=\;\sigma_{m_{i}}^{2}$, $\sigma_{i+N}^{2}=\sigma_{n_{i}}^{2}$, $\sigma_{i+2N}^{2}=\sigma_{v_{i}}^{2}$(i=1,⋯,N). The MLE to maximize the prior PDF is obtained by solving the following non-convex optimization problem.
(3)$$\hat{\theta }=\arg\min_{\theta }\;\sum_{i=1}^{3N}\frac{1}{\sigma_{i}^{2}}{\left[r_{i}-f_{i}\left(\theta \right)\right]}^{2}.$$

This estimator is evidently non-convex and difficult to solve, especially when both $P_{0}$ and β are unknown. In the next sections, we will derive a suboptimal technique to solve this problem.

### 2.3. Strategy for Parameter Estimation

In this section, we consider a general case where both transmit power $P_{0}$ and PLE β are unknown, then transform (Equation 3) to an alternative form that is easier to solve. For instance, when we need to estimate the target location of a mobile nodein an asynchronous system, it is often desirable to develop a distributed algorithm that is independent of any timing parameters. Furthermore, the system may need to be frequently recalibrated for optimal measuring. Thus, the joint estimate of the channel parameters (e.g., PLE, $P_{0}$), as well as the target location can be computed only from the received signals. We observe that if x is known, (Equation 3) becomes linear with respect to $P_{0}$ and β. Thus, if we obtain an estimate for the target location x, we can also obtain the estimates for $P_{0}$ and β. For a given maximum number of iterations *M*, the estimation procedures are described as follows.
Step 1: Set up initial guessing intervals for both $P_{0}$ and β, and pick ${\hat{P}}_{0}\in \left[P_{0\min},\;P_{0\max}\right]$ and $\hat{\beta }\in \left[\beta_{\min},\beta_{\max}\right]$.Step 2: Solve (Equation 3) using $\left({\hat{P}}_{0},\hat{\beta }\right)$ to obtain an estimate for $\hat{x}$.Step 3: Use ${\hat{x}}^{\left(k-1\right)}$ to obtain the linear solutions for $\left({\hat{P}}_{0}^{\left(k\right)},{\hat{\beta }}^{\left(k\right)}\right)$ by the following.
(4)$$\left({\hat{P}}_{0}^{\left(k\right)},{\hat{\beta }}^{\left(k\right)}\right)=\arg\min_{\left(P_{0},\beta \right)}\sum_{i=1}^{N}\frac{1}{\sigma_{i}^{2}}{\left[P_{i}-P_{0}+\beta t_{i}^{\left(k-1\right)}\right]}^{2},$$
where $t_{i}^{\left(k-1\right)}=10\log_{10}\left|\right|{\hat{x}}^{\left(k-1\right)}-a_{i}\left|\right|$. If either ${\hat{P}}_{0}^{\left(k\right)}$ or ${\hat{\beta }}^{\left(k\right)}$ does not lie in the predefined interval, i.e., ${\hat{P}}_{0}^{\left(k\right)}\notin \left[P_{0\min},\;P_{0\max}\right]$ or ${\hat{\beta }}^{\left(k\right)}\notin \left[\beta_{\min},\beta_{\max}\right]$, then stop the iteration. Otherwise, go to the next step.Step 4: Use the update $\left({\hat{P}}_{0}^{\left(k\right)},{\hat{\beta }}^{\left(k\right)}\right)$ to solve (Equation 3) again and get a new update for x, ${\hat{x}}^{\left(k\right)}$. Then, check the stopping criteria: stop if k>M or $|f\left(\theta^{\left(k\right)}\right)-f\left(\theta^{\left(k-1\right)}\right)|/f\left(\theta^{\left(k-1\right)}\right)\leq \epsilon$, otherwise, k=k+1, and go to Step 2.

### 2.4. Related Works

A typical wireless localization process follows three main step: range identification, location estimation, and location refinement. Due to hardware characteristics, ranging is not always available for wireless devices. Rather, it has been extracted from different characteristics of radio signals. Many localization algorithms utilized hybrid RSS-AoA measurements [14,15,19,20,21,22,23] because they are more accurate than other methods. The tradeoff is that using hybrid system provides more information on sensor location while increasing network complexity and implementation costs. Thus, according to a specific application purpose, we can select an appropriate localization scheme with a given level of complexity and accuracy.

There are some related literature works on hybrid RSS-AoA measurements [14,15,19,22,23,24]. According to the capabilities of diverse hardware, we may classify the conventional hybrid RSS-AoA-based localization into either of two main categories: nonlinear minimization localization or convex approximation localization. In the first category, supposing that the measurements from the reference node are only corrupted by zero-mean Gaussian noise and the reference nodes are identical or much closer to each other than the target, the basic idea is solving (3) with the given information. In particular, this nonlinear minimization problem can be solved by the Newton–Gauss iteration. For instance, ref. [19] proposed the least squares (LS) and optimization estimators using the distance and the AoA information. In [20], hybrid RSS-AoA LS and maximum likelihood (ML) estimators were derived for the emitter geolocation. The method of combining ML-LS implicitly assumes that a rough estimate range between anchor-target can be obtained. The cost function given by (Equation 3) weakly depends on this value; thus, the roughness will not significantly affect the solution. It has been reported in [22] that the accuracy of such a scheme suffers from the environmental dynamics. On the other hand, departing from the first category, most of the works in the second category simplify the location region update due to the target range and the network communication cost. The objective function (Equation 3) can be solved by itself or approximated by an alternative form. The approximation for (Equation 3) often makes use of the convex constraint. One typical example is [22] where the geometric constraints (distance and angle measurements) between target and anchor node are represented in linear inequalities. All of them are combined to form a single semidefinite programming (SDP) problem. For each iteration, the bounding region for each node is updated accordingly. In [14], the authors presented a hybrid weighted LS (WLS) approach to intra-cell target localization in non-line-of-sight environments, while another closed-form WLS was developed in [15]. The common characteristics of those approaches are their concise problem formulation with clear model representation and refined mathematical solutions, while likely precluding themselves in practice due to their complexities. For example, the complexity of solving the SDP in [22] is at least $O(N_{cv}^{3})$, where $N_{cv}$ is the number of convex constraints needed to describe the network connectivity and measurement information. Other approaches such as SDP and second-order cone programming were proposed in [22,23]. However, the aforementioned works were designed either on 2D space [14], or had a high computational complexity [22,23], or only worked well with low noise [14,19,24]. Moreover, the RSS signals mainly depend on the transmit power of the target node $P_{0}$ (e.g., its battery) and the PLE β. Only a few studies have considered the transmit power in (Equation 1) as an unknown parameter [15,20,22,23], while in practice, these two parameters may vary significantly in different times and places. To the best of our knowledge, there is no existing work that derives a solution for hybrid RSS-AoA in 3D space when both transmit power and PLE are unknown variables. Therefore, in the paper, we derive a suboptimal solution for this case.

Moreover, given the combined RSS-AoA measurements, our localization scheme is proposed based on the following assumptions. The first one is that the AoAs can be measured and calibrated from the omnidirectional antennas. The ambiguities in the hybrid receiver architecture and its sensitivity are neglected in this paper. We also do not address the noise uncertainty issue over fading channel factors through the localization problem formulation.

## 3. The Proposed Weighted Least Squares Approach

### 3.1. An Inspiring Example

In this section, we give an example for the system utilizing hybrid measurements as illustrated in Figure 2. In particular, we consider an intelligent vehicle parking system that consists of the following three parts: (i) a vehicle carries a mobile phone, which is able to connect with the Internet; (ii) a sensor system consists of several sensors placed at the parking slots to detect vehicle presence/absence and many anchor nodes and relay nodes located over the parking area. (iii) a base station receives the localization request message from the mobile phone and provides a possible parking slot. The relay node collects the sensor readings and transmits to the base station for further processing. The anchor node is placed at a known location and helps to cover the network connectivity. As shown in Figure 2a, a typical parking site is installed in several layers to optimize the parking space by saving on the depth and the height. Thus, it may have a lower maintenance and operational cost and a lower construction cost compared to the conventional ones. Each layer represents a parking level, which usually consists of three main components: anchor nodes, relay nodes, and sensors, as depicted in Figure 2b. Given this typical setting, a localization scheme is proposed based on two assumptions. The first one is that the layout of the parking system is organized in an optimal model, i.e., it satisfies the coverage, connectivity, and geolocation constraints, known as the optimal sensor placement model. Thus, the system is able to function effectively in terms of efficient data collection and financial cost. Second, the available of RSS holds over the whole network.

In order to apply the cooperative hybrid localization scheme, we performed two procedures: region identification and location refinement. The basic idea is that we first determined which area may have available slots by quantizing the RSS readings. Second, by utilizing the AoA measurements, we estimated the slot location. After that, a refinement method was applied for better accuracy.

### 3.2. Issues and Motivations

The aforementioned works [14,15,19,20,21,22,23] faced several issues. First, due to the nonlinear nature of the localization problem, the location estimation via the classical method such as the MLE requires high complexity and computation cost. On the other hand, low complexity approaches like the LS or WLS methods are based on the low noise assumption, which is hard to achieve in real-world applications. Thus, constructing a localization scheme that achieves very accurate positioning in a short time for both low and high noise conditions, as well as under different channel settings is needed. Second, it can be seen that the measurement noises have a large impact on the solution accuracy. Consequently, these effects must take part in the localization procedure. Thus, unlike the conventional approaches, we investigated the noise covariance of hybrid measurements and produced an unbiased estimate under a realistic assumption of unknown transmit power and PLE. Third, the current setting for the anchor nodes is seriously restricted to the ideal scenarios, that is the anchors are positioned at the vertices of a cube region in which the target is inside the convex hull formed by them; thus, no GDoP effect can be examined. For a more practical scenario, we considered a random network in the simulation setting, where the target may fall outside the convex hull defined by the anchors.

### 3.3. Quantized RSS-Based Ranging

We considered a realistic modification of (Equation 1), i.e., instead of using the analog version of the RSS measurements, we took the quantized values of them. In this case, we no longer have access to the actual RSS reading, but only partial information, related to the ability of the device to communicate with others. In particular, by using an *S*-level quantization operator Q(·), we obtain:(5)$$Q_{i}=Q\left(P_{i}\right)=\left\{\begin{array}{cc} 0, & L_{0}\leq P_{i}\leq L_{1}\\ 1, & L_{1}\leq P_{i}\leq L_{2}\\ \vdots & \;\;\;\vdots \\ S-1, & L_{S-1}\leq P_{i}\leq L_{S} \end{array}\right.,$$
where $L_{0},\cdots ,L_{S}$ are quantized levels, $L_{0}=-\infty$ dBm and $L_{S}=\infty$ dBm. Let ${\hat{d}}_{i}$ be the distance at which the mean received power is ${\bar{L}}_{i}$, i.e., the estimated distance between the target x and the anchor $a_{i}$; we can compute ${\hat{d}}_{i}$ from ${\bar{L}}_{i}$ as:(6)$${\hat{d}}_{i}=d_{0}10^{\frac{P_{0}-{\bar{L}}_{i}}{10\beta }}.$$

Figure 3 shows an example of distance estimation in indoor (PLE = 1.5 and additional reflection coefficients) and outdoor (PLE = 3.5) environments. Other parameters are given in Section 4. In each run, the estimated range was obtain from (Equation 6) and the quantization process. The coordinate of a single target was randomly generated for the given area. It has been shown in [3] that the relation between the average received power and the distance is proportional. The estimated range was obtained from the raw measurements over 100 realizations. We observed that in the outdoor environment the quantized scheme performed more stably and had smaller errors compared to the indoor space. This is because the wireless signals in the indoor environment suffered from multipath and reflection effects from walls and obstacles. In addition, the results provided a clue that could help in choosing an appropriate quantization level. A good accuracy of about 1 m average error within a range of 10 m can be achieved, which is acceptable for ranging with inexpensive devices.

### 3.4. Location Estimation

In the presence of both range and angle measurements, we can obtain the target location as:(7)$$\begin{array}{c} x=x_{i}+{\hat{d}}_{i}\sin{\hat{\alpha }}_{i}\cos{\hat{\phi }}_{i}\delta_{i},\\ y=y_{i}+{\hat{d}}_{i}\sin{\hat{\alpha }}_{i}\sin{\hat{\phi }}_{i}\delta_{i},\\ z=z_{i}+{\hat{d}}_{i}\cos{\hat{\alpha }}_{i}\delta_{i}, \end{array}$$
where $\delta_{i}$ is the bias-reducing constant for the RSS-AoA measurements given by:(8)$$\delta_{i}=\exp\left\{\frac{\sigma_{n_{i}}^{2}}{2}+\frac{\sigma_{v_{i}}^{2}}{2}-\frac{\sigma_{n_{i}}^{2}}{2{\left(\beta \gamma \right)}^{2}}\right\}.$$

Hence, (Equation 7) can be rewritten in a matrix form as $Ax+q=\hat{b}$, where $A\;=\;\mathrm{diag}\left\{e_{N},\;e_{N},\;e_{N}\right\}\;\in \;R^{3N\times 3},\hat{b}={\left[{\hat{b}}_{x},{\hat{b}}_{y},{\hat{b}}_{z}\right]}^{T}\in R^{3N\times 1}$ with three elements as:$$\begin{array}{cc} {\hat{b}}_{x} & ={\left[x_{1}+{\hat{d}}_{1}\sin{\hat{\alpha }}_{1}\cos{\hat{\phi }}_{1}\delta_{1},\cdots ,x_{N}+{\hat{d}}_{N}\sin{\hat{\alpha }}_{N}\cos{\hat{\phi }}_{N}\delta_{N}\right]}^{T},\\ {\hat{b}}_{y} & ={\left[y_{1}+{\hat{d}}_{1}\sin{\hat{\alpha }}_{1}\sin{\hat{\phi }}_{1}\delta_{1},\cdots ,y_{N}+{\hat{d}}_{N}\sin{\hat{\alpha }}_{N}\sin{\hat{\phi }}_{N}\delta_{N}\right]}^{T},\\ {\hat{b}}_{z} & ={\left[z_{1}+{\hat{d}}_{1}\cos{\hat{\alpha }}_{1}\delta_{1},\cdots ,z_{N}+{\hat{d}}_{N}\cos{\hat{\alpha }}_{N}\delta_{N}\right]}^{T}, \end{array}$$
and q is the noise vector, which has a zero mean vector and covariance $\Sigma_{x}=E\left[\left(\hat{b}-b\right){\left(\hat{b}-b\right)}^{T}\right]$. Thus, the WLS solution is obtained by minimizing the following cost function.
(9)$$\epsilon \left(x\right)={\left(\hat{b}-Ax\right)}^{T}\Sigma_{x}^{-1}\left(b-Ax\right),$$
which yields the intermediate solution:(10)$$\hat{x}=A^{†}b^{†},$$
where $A^{†}={\left[A^{T}\Sigma_{\hat{x}}^{-1}A\right]}^{-1}A^{T}$ and $b^{†}=\Sigma_{\hat{x}}^{-1}\hat{b}$.

### 3.5. Linear Approximation and Channel Factor Refinement

From Section 3.4, the target node should obtain its initial location estimate. In some cases, it might not be well localized since it does not have enough anchor nodes. In order to have knowledge on the location estimate and internal relay parameters of its neighbors, the location estimate of the target is refined by applying a simple LS algorithm. Thus, the square root term in (Equation 6) can be linearized by using the Taylor series approximation. Following Section 2.3, first we pick initial ${\hat{P}}_{0}\in \left[P_{0\min},\;P_{0\max}\right]$ and $\hat{\beta }\in \left[\beta_{\min},\beta_{\max}\right]$. The right-handed side of (Equation 6) can be approximated using the first-order Taylor expansion as:(11)$$\begin{array}{cc} {\hat{d}}_{i} & =\sqrt{{\left(x-x_{i}\right)}^{2}+{\left(y-y_{i}\right)}^{2}+{\left(z-z_{i}\right)}^{2}}\\ & \approx {\tilde{d}}_{i}+\frac{\partial {\hat{d}}_{i}}{\partial x}\delta x+\frac{\partial {\hat{d}}_{i}}{\partial y}\delta y+\frac{\partial {\hat{d}}_{i}}{\partial z}\delta z\\ & \approx {\tilde{d}}_{i}+\frac{\tilde{x}-x_{i}}{{\tilde{d}}_{i}}\left(x-\tilde{x}\right)+\frac{\tilde{y}-y_{i}}{{\tilde{d}}_{i}}\left(y-\tilde{y}\right)+\frac{\tilde{z}-z_{i}}{{\tilde{d}}_{i}}\left(z-\tilde{z}\right). \end{array}$$

Here, ${\left(x_{i},y_{i},z_{i}\right)}^{T}$ is the known coordinate of either an anchor or a known neighbor node and ${\tilde{d}}_{i}\;=\;\sqrt{{\left(\tilde{x}-x_{i}\right)}^{2}+{\left(\tilde{y}-y_{i}\right)}^{2}+{\left(\tilde{x}-z_{i}\right)}^{2}}$. Thus, the location estimate is updated by $\hat{x}\leftarrow \hat{x}+\delta x$, $\hat{y}\leftarrow \;\hat{y}+\delta y$ and $\hat{z}\leftarrow \hat{z}+\delta z$.

Next, the unknown PLE can be expressed as $\hat{\beta }=\beta_{0}\left[1+\frac{\hat{\beta }-\beta_{0}}{\beta_{0}}\right]=\beta_{0}\left[1+\delta \right]$, where $\beta_{0}\neq 0$ is the tuning parameter, which can be set in the simulation, and $\delta =\frac{\beta_{0}-\hat{\beta }}{\beta_{0}}$. For small |δ| and small $\frac{{\hat{P}}_{0}-P_{i}}{10\beta_{0}}\delta \ln10$, by using the facts that 1/(1+δ)≈1−δ and $u^{-\delta }\approx 1-\delta \ln u$, let $\lambda_{i}=10^{\frac{{\hat{P}}_{0}-P_{i}}{10\beta_{0}}}$, and we perform:(12)$$\begin{array}{cc} {\hat{d}}_{i} & \approx 10^{\frac{{\hat{P}}_{0}-P_{i}}{10\beta_{0}\left(1+\delta \right)}}\eta \left[1+\frac{\ln10}{10\hat{\beta }}m_{i}\right]\\ & \approx \lambda_{i}\eta \left(1-\delta \ln\lambda_{i}\right)\left[1+\frac{\ln10}{10\hat{\beta }}m_{i}\right], \end{array}$$
where $\eta ≜10^{\left(P_{0}-{\hat{P}}_{0}\right)/\left(10\hat{\beta }\right)}$. In order to take advantage of neighboring anchor nodes, each weight factor is defined as $w_{i}=1-P_{i}/\left(\sum_{i=1}^{N}P_{i}\right)$.

In the next step, we try to extract the channel factors from the raw sensor data based on (Equation 4). By denoting ${\bar{d}}_{i}=\left|\right|\bar{x}-a_{i}\left|\right|$ (i.e., the new estimated distance between the target node and the ith anchor node), we first rewrite the path-loss model as:(13)$$P_{i}=P_{0}-10\beta \log_{10}{\bar{d}}_{i}+m_{i}.$$
Here, an LS method is applied to compute the estimates of the transmit power $P_{0}$ and the PLE β. The solution is obtained by:(14)$$\left[{\bar{P}}_{0},\;\bar{\beta }\right]={\left(A_{1}^{T}A_{1}\right)}^{-1}A_{1}^{T}b_{1},$$
where $A_{1}=\left[\begin{array}{cc} 1 & -10\log_{10}{\bar{d}}_{1}\\ \vdots & \vdots \\ 1 & -10\log_{10}{\bar{d}}_{N} \end{array}\right],b_{1}={\left[P_{1},\cdots ,P_{N}\right]}^{T}$. Then, we exploit $\left[{\bar{P}}_{0},\;\bar{\beta }\right]$ to calculate the distance (Equation 12) and use its estimated value to solve the WLS problem given by (Equation 10). Note that in Step 1 given in Section 2.3, the initialization state sets up the guessing interval of channel factors based on the mean RSS data between a reference node and anchors in the environment.

**Remark** **1**(Complexity analysis). *Given a maximum number of iterations M, the complexity of solving MLE, LS, WLS, SOCP, mixed SDP/SOCP, and the proposed method is summarized in Table 1.*

### 3.6. Cramer–Rao Lower Bound for the Proposed Hybrid RSS-AoA Localization

In order to provide a limit of estimator performance, we computed the CRLB to reflect the effect of model parameters on the estimation accuracy. Let $\hat{\theta }$ be an unbiased estimator of a parameter θ; the CRLB for the ith element of $\hat{\theta }$ is defined by:(15)$$\mathrm{Var}\left({\hat{\theta }}_{i}\right)={\left[J^{-1}\left(\theta \right)\right]}_{i,i}≜\mathrm{CRLB}\left({\hat{\theta }}_{i}\right).$$

Let p(r|θ) denote the conditional PDF of the observation data $r={\left[P^{T},\phi^{T},\alpha^{T}\right]}^{T}$ parameterized by θ. The entries of the corresponding Fisher information matrix J(θ) are derived as:(16)$${\left[J\left(x\right)\right]}_{i,j}=-E\left[\frac{\partial \ln p(r|x)}{\partial x_{i}}\frac{\partial \ln p(r|x)}{\partial x_{j}}\right].$$

When applying the quantization operation (Equation 5), the conditional PDF of $r_{i}=P_{i}\in \left[L_{s},L_{s-1}\right]$ (i=1,⋯,N) is given by:(17)$$p\left(r_{i}|\theta \right)=\int_{L_{s}}^{L_{s-1}}p_{i}\left(L_{i}\right)dP=\Phi \left(D_{i}^{s-1}-D_{i}^{s}\right),$$
where $\Phi \left(t\right)=\int_{-\infty }^{t}\frac{1}{\sqrt{2\pi }}e^{-v^{2}/2}dv,\Phi \left(D_{i}^{0}\right)=1,\Phi \left(D_{i}^{S}\right)=0$, and $D_{i}^{s}=\frac{\beta }{\sigma_{m_{i}}^{2}}\frac{\ln(d_{i}/{\hat{d}}_{i})}{\ln(d_{i}/d_{0})},i=1,\cdots ,N$.

Let $d_{2,i}=\sqrt{{\left(x-x_{i}\right)}^{2}+{\left(y-y_{i}\right)}^{2}}$, then we have by some manipulations,
(18)$$\begin{array}{c} \frac{\partial f_{i}\left(x\right)}{\partial x}={\left[\frac{10\tau_{i}\left(x-x_{i}\right)}{d_{i}^{2}\ln10},-\frac{\left(y-y_{i}\right)}{d_{2,i}^{2}},\frac{\left(x-x_{i}\right)\left(z-z_{i}\right)}{d_{i}^{2}d_{2,i}}\right]}^{T}, \end{array}$$
(19)$$\begin{array}{c} \frac{\partial f_{i}\left(x\right)}{\partial y}={\left[\frac{10\tau_{i}\left(y-y_{i}\right)}{d_{i}^{2}\ln10},\frac{\left(x-x_{i}\right)}{d_{2,i}^{2}},\frac{\left(y-y_{i}\right)\left(z-z_{i}\right)}{d_{i}^{2}d_{2,i}}\right]}^{T}, \end{array}$$
(20)$$\begin{array}{c} \frac{\partial f_{i}\left(x\right)}{\partial z}={\left[\frac{10\tau_{i}\left(z-z_{i}\right)}{d_{i}^{2}\ln10},0,-\frac{d_{2,i}}{d_{i}^{2}}\right]}^{T}, \end{array}$$
(21)$$\begin{array}{c} \frac{\partial f_{i}\left(x\right)}{\partial P_{0}}={\left[1,0,0\right]}^{T},\frac{\partial f_{i}\left(x\right)}{\partial \beta }={\left[-10\log_{10}d_{i},0,0\right]}^{T}, \end{array}$$
where:(22)$$\tau_{i}=\frac{1}{2\pi }\sum_{i=1}^{S}\frac{\left[e^{-\frac{1}{2}{\left(D_{i}^{s-1}\right)}^{2}}\left(\frac{\beta }{\sigma_{m_{i}}}-D_{i}^{s}\right)-e^{-\frac{1}{2}{\left(D_{i}^{s}\right)}^{2}}\left(\frac{\beta }{\sigma_{m_{i}}}-D_{i}^{s}\right)\right]}{\Phi \left(D_{i}^{s-1}\right)-\Phi \left(D_{i}^{s}\right)}.$$

By using the Taylor series, we construct the relationship between x and $\hat{x}$ as:(23)$$x\approx \hat{x}+\sum_{i=1}^{3N}\frac{\partial x}{\partial f_{i}}\left[r_{i}-f_{i}\left(x\right)\right],$$
where $\frac{\partial x}{\partial u}={\left(A^{T}WA\right)}^{-1}A^{T}W\frac{\partial b}{\partial u}$. Assuming that the measurements are independent and differentiable, the approximated variances of the estimator errors are given by:(24)$$\begin{array}{cc} \mathrm{Var}\left({\left[\hat{x}\right]}_{j}\right)\approx \sum_{i=1}^{N} & \left\{{\left({\left[\frac{\partial x}{\partial P_{i}}\right]}_{j}\right)}^{2}\sigma_{m_{i}}^{2}+{\left({\left[\frac{\partial x}{\partial \phi_{i}}\right]}_{j}\right)}^{2}\sigma_{n_{i}}^{2}+{\left({\left[\frac{\partial x}{\partial \alpha_{i}}\right]}_{j}\right)}^{2}\sigma_{v_{i}}^{2}\right\},\;j=1,2,3. \end{array}$$

## 4. Simulation Results

Computer simulations were conducted to show the performance of the proposed approach compared to the MLE (solving by the Gauss–Newton method [25]), the SOCP (using the CVX package [26]), and the WLS method [21]. Simulation parameters have been adopted, and we kept the same set to achieve fairness of comparison among the aforementioned works. Furthermore, the true values of the unknown parameters were used as the starting points. In particular, the following parameters were used: $d_{0}=1$ m, $P_{0}∼\mathrm{Unif}\left[-15,-5\right]$ dBm, β∼Unif[2,5], the standard deviationof the RSS was $\sigma_{m_{i}}=\sigma_{1}$, and that of the AoA was $\sigma_{n_{i}}=\sigma_{v_{i}}=\sigma_{2}$. We used *N* anchors to locate one unknown sensor node. Those anchor nodes were picked randomly in a cube-shaped region, which had an edge length of 100 m. The unknown node was deployed randomly in this region. In terms of sensor deployment settings, the distance between the normal nodes and the anchor nodes should meet the connectivity condition of the networks. The maximum communication among nodes was set at 25 m. We used (Equation 1) to generate hybrid RSS-AoA measurements.

In order to evaluate the performance, the root mean squared error (RMSE) and the bias were considered as follows.
(25)$$\begin{array}{c} \mathrm{RMSE}\left(\hat{x}\right)=\sqrt{\frac{1}{M_{c}}\sum_{m=1}^{M_{c}}\left|\right|x^{\left(m\right)}-{\hat{x}}^{\left(m\right)}{\left|\right|}^{2}}, \end{array}$$
(26)$$\begin{array}{c} \mathrm{Bias}\left(\hat{x}\right)=\frac{1}{M_{c}}\sum_{m=1}^{M_{c}}\left|\right|x^{\left(m\right)}-{\hat{x}}^{\left(m\right)}{\left|\right|}_{1}, \end{array}$$
where $M_{c}=5000$ is the number of Monte Carlo runs. In fact, small RMSE and bias values can produce the estimated solutions that are close to the traceable actual values. Given a localization algorithm, the RMSE and the bias show how well the estimated target location matches the actual one. More specifically, besides the computation cost, a good localization algorithm must ensure a low RMSE and a stable bias under the same settings scenario. In this section, we will evaluate and compare those aforementioned algorithms under different noise conditions to make accurate validation.

In order to first find an appropriate threshold for the RSS measurements, we investigated the effect of the RSS quantization level on the RMSE performance. Without loss of generality, we set $\sigma_{1}=\sigma_{2}^{2}$ and N=10. One of the basic choices in selecting the quantization level is the number of anchors. As Figure 4 illustrated, the higher quantized level *S* resulted in a lower RMSE value. However, the RMSE performance was not improved by increasing the quantization level *S*. This is because when *S* becomes large, there is not much difference among the quantized intervals, and boosting error can increase with *S*, especially when the measurements are noisy. We also observed that the value of quantized level had a significant impact on the performance of the estimated range (Equation 6). A small value *S* reduced the localization expense, but degraded the localization accuracy. When *S* approached infinity, the estimator in (Equation 10) was actually reduced to the conventional weight least squares method with the bias-reducing constant (Equation 8). Thus, with a refinement scheme to exploit the above tradeoff, the proposed localization scheme worked well under the condition settings. With an acceptable RMSE, we chose S=8 for the next simulation results.

In Figure 5, we evaluate the RMSE with different numbers of anchor nodes *N* where $\sigma_{1}^{2}=\sigma_{2}^{2}=0$ (i.e., noiseless measurements). When the number of anchor nodes *N* increased, the RMSEs of all estimators decreased significantly and converged to the CRLB. With a fixed number of anchor nodes, the proposed estimator achieved an acceptable RMSE and smaller bias compared to the others.

Figure 6 illustrates the RMSE and the bias versus $\sigma_{1}$ when $\sigma_{2}=1$ deg and N=13. We observed that the proposed scheme achieved a comparable RMSE and smaller bias when $\sigma_{1}$ was small compared to other methods. When $\sigma_{1}$ became large, even our solution had a lower RMSE, but higher bias. This phenomenon may occur from the approximations of the linearization process explained in Section 3.5.

Figure 7 shows the RMSE and the bias versus $\sigma_{2}$ when $\sigma_{1}=3$ dBm. From this figure, we observe that the proposed method still had a good performance when the noise became large. The bias performance was comparable to the MLE and SOCP and much lower than the WLS. However, it was also acceptable since the proposed method had a lower complexity than the MLE and SOCP estimator.

**Remark** **2.**
*In some approaches, the accuracy may be defined as the expected distance between the actual position x and the estimated one $\hat{x}$, i.e., $E(||\hat{x}-x||)$. Thus, the evaluation metric can be indicated as the percentage of the results satisfying a predefined accuracy requirement. In this section, we only considered two metrics in (Equation 25) and (Equation 26) to validate the localization effectiveness. Based on the observation from the simulation results, there were some major factors that caused large errors during the localization process. Assuming that information on the anchor positions was obtained accurately and reliably, the error source may be caused by the distance estimation error from (Equation 6), the multipath fading, and the AoA measurement noises. The same phenomenon occurred in other localization schemes.*


**Remark** **3.**
*It has been seen that that localization system runs based on an event-driven mechanism, i.e., a sensor only collects the RSS-AoA measurements when it receives a localization request. The quantized RSS helps the sensors to know quickly their ranges to the neighbor anchors/relay nodes and to indicate the AoA of the incoming signals. Until the localization packet (sensor ID, RSS-AoA measurements) reaches the neighbor anchor nodes, they collect and deliver back to the BS their locations. Finally, the localization algorithm can be executed by the BS to obtain the sensor location on the received packets. The unknown node only sends one message for localization instead of continuously reporting the measurements to the neighbor anchors; thus, the number of messages required for localization per unknown node can be reduced.*


In order to measure the uncertainty of the location estimate $\hat{x}$ relative to its mean $E(\hat{x})$, we considered the GDoP that measured the ratio of the RMSE of a location fix to the statistical RMSE of the ranging measurements as [3]: (27)$$\mathrm{GDoP}\left(\hat{x}\right)=\frac{1}{\sigma_{r}}\sqrt{E\left[{\left\{\hat{x}-E\left(\hat{x}\right)\right\}}^{T}\left\{\hat{x}-E\left(\hat{x}\right)\right\}\right]}.$$

We now investigate the effects of the GDoP in the network localization. For a given bound ξ, we calculated the probability of GDoP smaller than ξ, which means the probability of acceptable estimation $P\{\hat{x}:\mathrm{GDoP}\left(\hat{x}\right)\leq \xi \}$. The exponential increase in the detection probability as we took more measurements from the anchor nodes is illustrated in Figure 8, which plots the performance predicted by (Equation 27) for a range of ξ with the number of anchor nodes *N*. The results are consistent and should approach one exponentially fast as *N* and ξ are increased.

## 5. Conclusions

In this paper, we introduced the localization problem on RSS-AoA measurements in 3D WSNs, then derived a suboptimal solution that can obtain a good accuracy under some mild conditions. Simulation results showed that the proposed method outperformed the MLE, SOCP, and WLS solutions in terms of the RMSE and the complexity and had an acceptable bias performance.

Some effectiveness of the proposed scheme is addressed as follows. First, as we presented in Section 2.4 and Section 3.2, we considered two main problems that had not been solved yet: (i) how to handle the nonconvexity of network deployment with low computational complexity; (ii) how to extract quickly the location information to keep lower maintenance. It is not easy to solve both problems in 2D space, but much more difficult in 3D space. Moreover, since the conventional approaches were often conducted in a distributed or an iterative manner, error during estimation process leads to inaccurate location estimates. Through mathematical analysis and simulation, the proposed approach has demonstrated that it is an effective technique to tackle those issues. Second, with the event-based localization protocol, a sensor node can save its energy and does not require to wake up all the time. Thus, the proposed scheme is suitable for location-aware applications.

In the future work, we are interested in device-free localization algorithms where the “to-be-located target” cannot join the online localization process. It needs to be detected and localized by using the changes of the environment (e.g., RSS changes) collected at scattered devices. Another direction of future research with good potential is investigating different radio propagation under multipath, shadowing affects caused by reflection and scattering from obstacles, and interference with other devices that operate at the same frequency, etc. The location of unknown nodes cannot be uniquely determined due to ranging errors from location ambiguities in a practical scenario. Moreover, the newly-estimated locations may become the references to help with locating other sensor nodes in the localization process for the multiple target localization problem. 

## Figures and Tables

**Figure 1 sensors-19-02121-f001:**
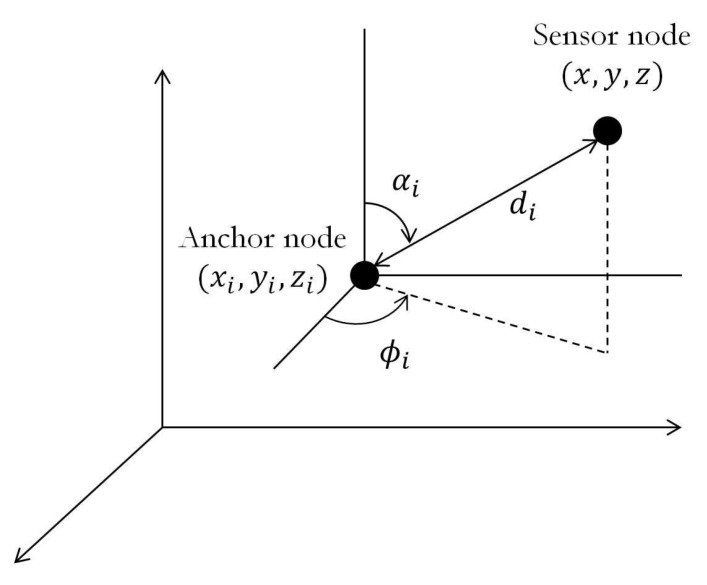
Illustration of hybrid RSS-AoA localization in 3D space.

**Figure 2 sensors-19-02121-f002:**
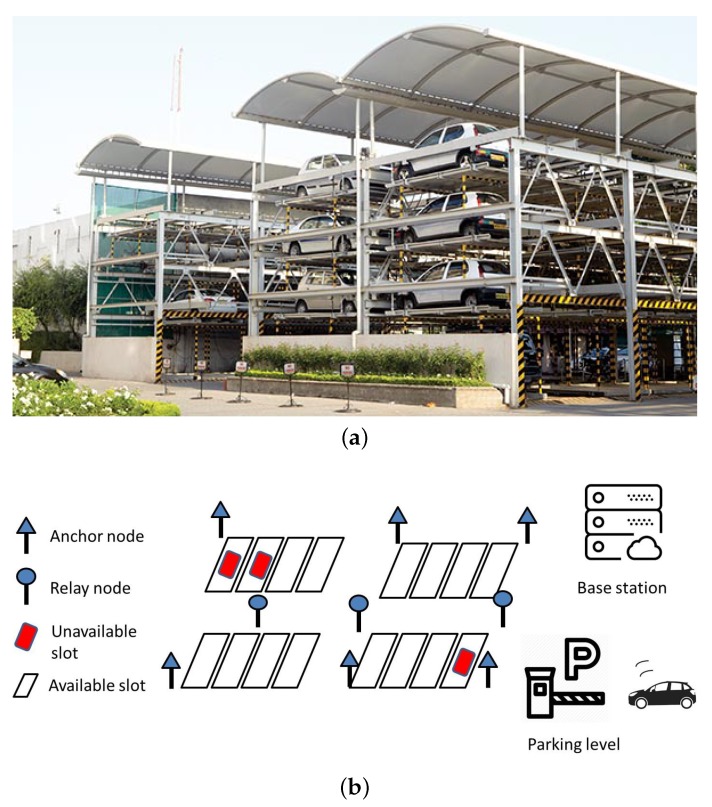
Example of an intelligent vehicle parking system. (**a**) A typical multi-level parking site [24]; (**b**) Infrastructure at a single layer.

**Figure 3 sensors-19-02121-f003:**
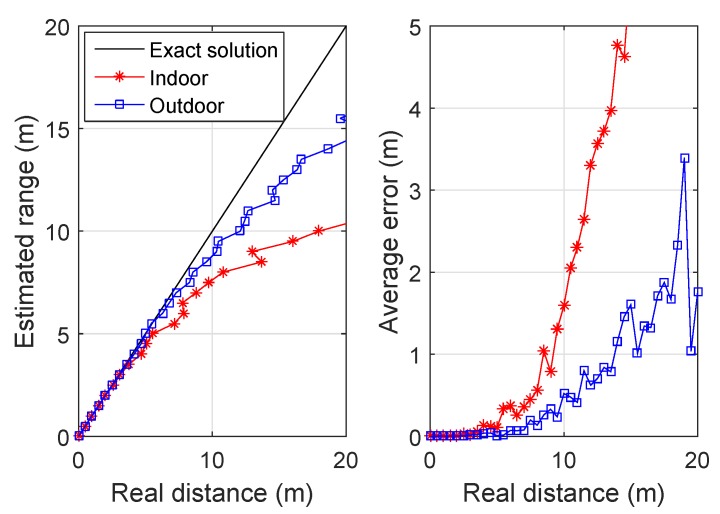
An example of the quantized RSS scheme when S=10.

**Figure 4 sensors-19-02121-f004:**
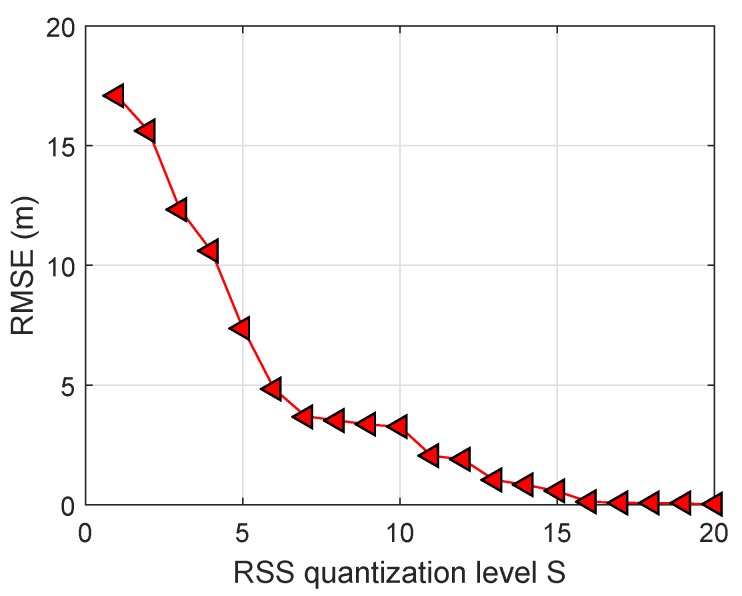
Impact of RSS quantization levels on root mean squared error (RMSE).

**Figure 5 sensors-19-02121-f005:**
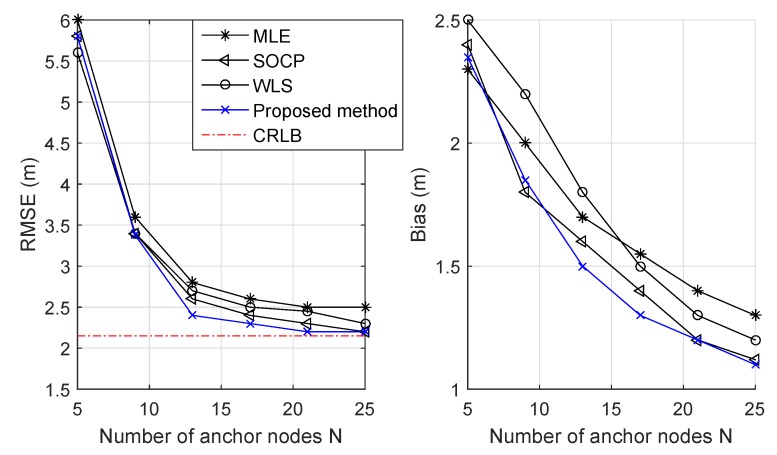
RMSE versus number of anchor nodes *N*.

**Figure 6 sensors-19-02121-f006:**
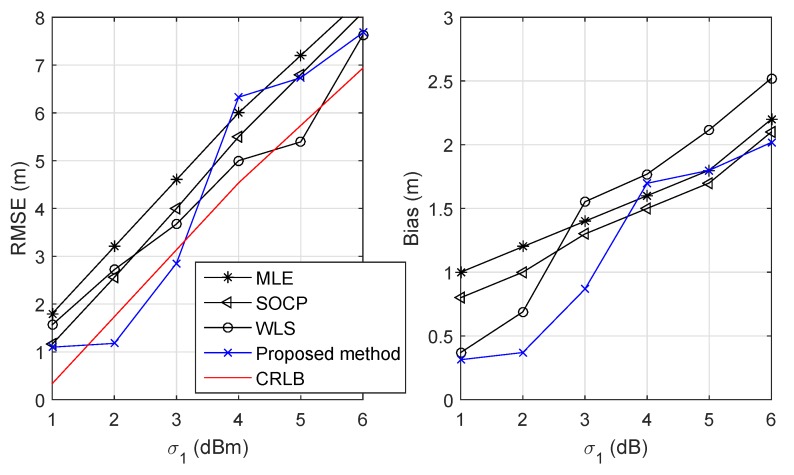
RMSE and bias versus standard deviation of the RSS measurement errors $\sigma_{1}$.

**Figure 7 sensors-19-02121-f007:**
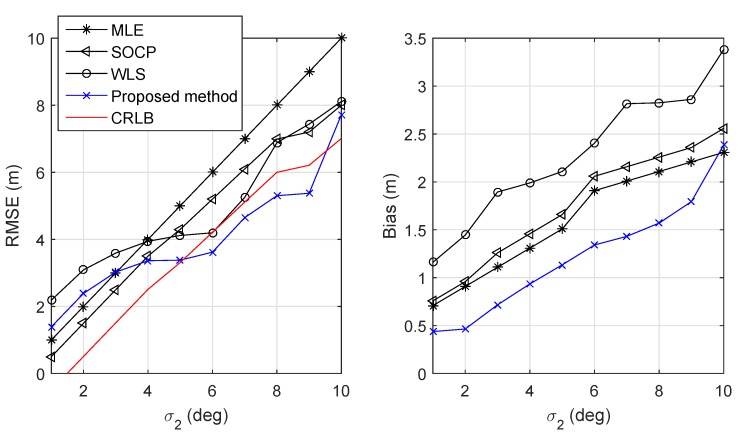
RMSE and bias versus standard deviation of AoA measurement errors $\sigma_{2}$.

**Figure 8 sensors-19-02121-f008:**
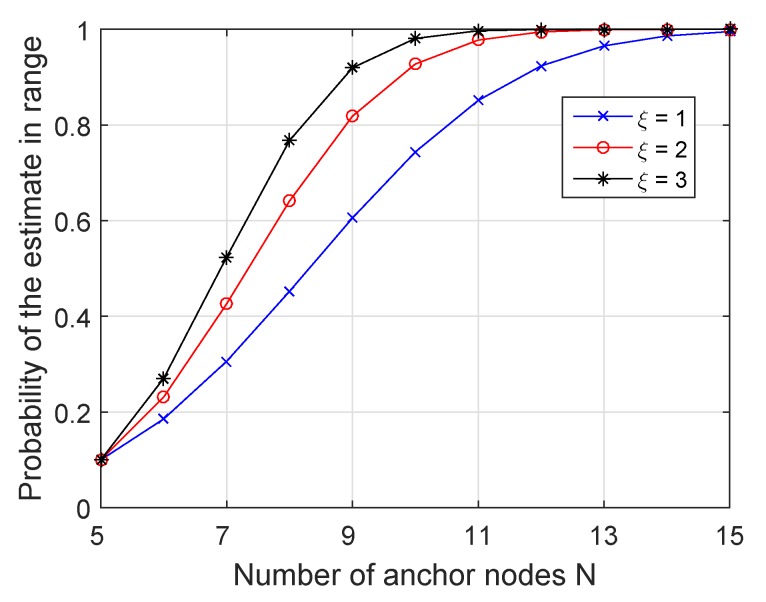
GDoP versus number of anchor nodes *N*.

**Table 1 sensors-19-02121-t001:** Complexity of different approaches.

Method	Complexity
MLE [20]	$O(M*N^{3})$
ine LS [19]	$O(K_{\max}N)$, where $K_{\max}$ is the maximum number of steps in the bisection procedure
ine WLS [14]	O(MN+KN), where *K* is the maximum number of steps in the bisection procedure [25]
ine SOCP [21]	$O(M_{SOCP}*{\left(N+4\right)}^{3.5})$, where $M_{SOCP}$ is related to a search interval [p,q] and the solution precision ϵ: $M_{SOCP}$ is the minimum integer that satisfies $M>\log_{2}\left(\left(q-p\right)/\epsilon \right)$.
ine Mixed SDP/SOCP [23]	At least $O(\sqrt{3}\left(81{\left(N+0.5\right)}^{2}\right))$
ine Proposed method	$O(MN+K_{\max}N)$, where $K_{\max}$ is the maximum number of iterations for solving (Equation 14).

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
