# Peer review of "An Efficient Hybrid RSS-AoA Localization for 3D Wireless Sensor Networks"

_sensors, 2019, doi:10.3390/s19092121_

Round 1
Reviewer 1 Report
The paper is well written and the research analysis with the corresponding findings and results are clearly presented. The major issue is the paper novelty, as the main contribution of the paper is in joint RSS and AOA based localization, which is rather well investigated. However, this is also indicated by the authors, who provide a good list of references from the same topic. Hence, unfortunately, the novelty of the paper is at low level and the presented approach with WLS is rather straightforward and predictable. The presented location likelihood function can be solved, for example, with conventional non-linear least squares methods, which would probably give better result than the proposed approach, if they were properly configured. Of course, authors present results with Gauss-Newton method, but it remains unclear, how the algorithm is exactly implemented.
In addition, there is one technical question, which is left unclear: Why CRLB shown in Fig. 5 does not decrease when the number of anchor nodes increases?
Author Response
Please find the attached file for detail.

Reviewer 2 Report
The method proposed in this paper overcomes the non-convexity of maximum likelihood estimation method. In logistic regression, the maximum likelihood estimation is used as a loss function because of the non-convexity of least squares. The non-convexity of the maximum likelihood estimation method corresponding to this hybrid RSS-AoA localization for 3D problem need to be demonstrated in detail.
RSS is easy to be measured but measuring AoA requires an antenna array, the hardware configuration and cost of each anchor node would be introduced for readers to reference.
L0=-∞ and LS=∞, the actual measurement range of RSS is much smaller. As an approximate processing, can it narrow the range and improve the accuracy? The geometric meaning of Quantized RSS-Based Ranging is illustrated by a diagram.
Author Response
Please find the attached file for detail

Reviewer 3 Report
The authors proposed a weighted LS for localization estimation, in particular, while transmit power and the path loss exponent unknown. Still, there are some issues needed to address to improve the quality of the paper up to the standard as examples below.
1. The authors should revise the organization to ease the reader understanding; the problem, motivation, what others’ limitations, what is the techniques (novel or application), simulation, comparative results, justification, conclusion/future, etc.
2. Please provide a comprehensive state of arts for any claims and assumptions as the citations.
3. Note that while claiming others may have low precision or high computation, recent state of the arts can mitigate those and so please provide some justification and details.
4. Please provide some cost analysis for the proposed one.
5. Please provide some justification over the applicable of RSS-AOA, say, with limited resource on sensors but with antenna array? Also, what about the precision of the way to map RSS to distance.
6. Please be specific how to overcome the two limitations of RSS-AoA based approach here?
7. Please revise the motivation leading to the contribution (not clear), in particular, with unknown PLE.
8. Please justify the contribution a bit against the previous paper by the same authors [14].
9. Please justify “fast” scheme say that depends on many factors/setup/testbed OR the author meant low complexity?
10. Related work should be comprehensive and intensive study.
11. In terms of derivation and analysis, with one additional axis say z, what makes it so different from the 2D analysis?
12. If correctly understand, there is no “x” in (3).
13. What is “initial guessing intervals”?
14. How to define the range say min and max; and any sensitivity analysis?
15. The evaluation shouldn’t be contribution; there is a need to perform the evaluation to state the superiority of the algorithm.
16. It’s not quite clear why the assumption is over just 1 unknown node OR this is just the formation and further, the model should not limit the #unknown nodes.
17. Since the authors claimed on the real-environment testbed; details of all related setups are required including detailed characteristics of each component. Also, any specific protocol used in this model?
18. Whether is any coverage and routing used here? It’s a bit confusing when the authors stated the real example applied in this proposal.
19. Based on Table 1, the complexity (most) is O of N, isn’t it? Please be specific on the superiority of the proposed scheme.
20. The authors claimed on “high computational cost” and “measurement errors”; please provide some analysis against the state of the arts.
21. The authors should perform the comparative analysis against the state of the arts.
22. Detailed configuration/setup/testbed are required in the evaluation section.
23. Please provide some discussion on the performance; say, it seems there is not much trend on the RMSE over all; very close to each other.
24. The authors claimed “..problem with the constraints in energy consumption and communication bandwidth.”; please provide some analysis on the latter (comm. BW).
Author Response

(The authors gave the same response as above.)

Round 2
Reviewer 3 Report
Please make it more clear on such comments below.
5. Please provide some justification over the applicable of RSS-AOA, say, with limited resource on sensors but with antenna array? Also, what about the precision of the way to map RSS to distance.
Comments: Still not clear as an example of sensor networks; please provide an example of sensor (mote?) embedded with multi antenna array; also, please provide some details for sensor networks here, say, in terms of communication aspect – 802.15?. Also, please, if any, provide some citations on the RSS to distance mapping technique.
18. Since the authors claimed on the real-environment testbed; details of all related setups are required including detailed characteristics of each component. Also, any specific protocol used in this model?
Comments: Please provide some sort of communication protocol or even with some valid assumption. Also, what about the communication overhead? Any kind of messages sending between sensors?
19. Based on Table 1, the complexity (most) is O of N, isn’t it? Please be specific on the superiority of the proposed scheme.
Comments: Still not clear; most = O(N); also please do analysis over [22].
20. Please provide some discussion on the performance; say, it seems there is not much trend on the RMSE over all; very close to each other.
Comments: Still not clear; seems the results are close to each other say how much it can be improved in what factor? Is it really statistically impact? Also, please provide CI say 95%.
BTW: The authors should provide detailed simulations say parameters and configurations; for example, testbed, communication protocol, sensor technology, etc. Also, different set of scenario should be investigated like the impact of #nodes (density), area – small/large, the effect of obstacle if any, etc., else it’s difficult to see the contribution in terms of “WSN” application.
Author Response

(The authors gave the same response as above.)
